# Multi-Scale FPGA-Based Infrared Image Enhancement by Using RGF and CLAHE

**DOI:** 10.3390/s23198101

**Published:** 2023-09-27

**Authors:** Jialong Liu, Xichuan Zhou, Zhenlong Wan, Xuefei Yang, Wei He, Rulong He, Yingcheng Lin

**Affiliations:** 1The School of Microelectronics and Communication Engineering, Chongqing University, Chongqing 400044, China; liujialong6161@gmail.com (J.L.); zxc@cqu.edu.cn (X.Z.); 202212131168@stu.cqu.edu.cn (X.Y.); hewei007@cqu.edu.cn (W.H.); 0909031043@nue.edu.cn (R.H.); 2National Information Center of GACC, Beijing 100010, China; wanzhenlong@mail.customs.gov.cn; 3The School of Electronic Engineering, Naval University of Engineering, Wuhan 430030, China

**Keywords:** infrared image enhancement, multi-scale decomposition, real-time, FPGA

## Abstract

Infrared sensors capture thermal radiation emitted by objects. They can operate in all weather conditions and are thus employed in fields such as military surveillance, autonomous driving, and medical diagnostics. However, infrared imagery poses challenges such as low contrast and indistinct textures due to the long wavelength of infrared radiation and susceptibility to interference. In addition, complex enhancement algorithms make real-time processing challenging. To address these problems and improve visual quality, in this paper, we propose a multi-scale FPGA-based method for real-time enhancement of infrared images by using rolling guidance filter (RGF) and contrast-limited adaptive histogram equalization (CLAHE). Specifically, the original image is first decomposed into various scales of detail layers and a base layer using RGF. Secondly, we fuse detail layers of diverse scales, then enhance the detail information by using gain coefficients and employ CLAHE to improve the contrast of the base layer. Thirdly, we fuse the detail layers and base layer to obtain the image with global details of the input image. Finally, the proposed algorithm is implemented on an FPGA using advanced high-level synthesis tools. Comprehensive testing of our proposed method on the AXU15EG board demonstrates its effectiveness in significantly improving image contrast and enhancing detail information. At the same time, real-time enhancement at a speed of 147 FPS is achieved for infrared images with a resolution of 640 × 480.

## 1. Introduction

Visible light sensors can capture high-resolution images with rich textures and detailed information. However, the image quality captured by visible light sensors is greatly affected by the light environment. Poor illumination can degrade the visual image’s quality, leading to issues such as glare, smoke, and overexposure. In contrast, infrared imaging technology utilizes differences in infrared radiation intensity for object detection, thereby making it less susceptible to varying lighting and weather conditions [1]. Therefore, infrared image enhancement has become a hot topic in current research and has the potential to bring significant benefits to the field of multi-modal information fusion [2,3,4,5]. Simultaneously, it is extensively employed in fields such as target detection [6,7,8] and medical diagnostics [9,10]. However, atmospheric attenuation, scattering, and refraction can introduce noise in infrared images, resulting in low contrast, reduced signal-to-noise ratio, and blurred details, which greatly affect the detection, recognition, and infrared tracking of targets [11,12]. To satisfy the requirements for practical applications, effective image enhancement algorithms that improve contrast, reduce noise, and address the problem of detail blurring caused by interference must be employed. Infrared image enhancement algorithms can be broadly categorized into three main domains: spatial domain, frequency domain, and convolutional neural networks (CNNs).

Spatial domain enhancement algorithms are primarily based on histogram equalization (HE). HE [13] is widely employed to enhance the contrast of infrared images. It alters the distribution of gray values in the original image according to the frequency of each gray level in the image histogram to achieve a uniform distribution. However, because HE is applied to the entire image globally, it may result in noise enhancement and the weakening of image details, thus affecting image quality [14]. In recent years, some improved HE-based algorithms have been proposed, such as brightness-preserving bi-histogram equalization (BPBHE) [15], dualistic sub-image histogram equalization (DSIHE) [16], and minimum mean brightness error bi-histogram equalization (MMBEBHE) [17]. BPBHE divides the histogram of the image into two sub-histograms around the average gray value of the image and then independently equalizes each part to preserve brightness, thus overcoming the challenge of maintaining the same level of brightness throughout the image. DSIHE employs a similar process as BPBHE, although it separates the histogram by median instead of mean. However, DSIHE is suitable only for images that exhibit uniform intensity distribution and has limited effectiveness in preserving the original brightness. MMBEBHE is a variant of BPBHE. It first separates the histogram by using a designated threshold that maintains the minimum mean brightness error between the input and output images. Next, it independently equalizes the two parts. MMBEBHE is an improvement over BPBHE and DSIHE; however, it has limitations in terms of preserving contrast and brightness. Recursive mean-separate histogram equalization [18] and recursive sub-image histogram equalization [19] are the recursive versions of BPBHE and DSIHE, respectively. They provide a flexible way of monitoring the degree of over-enhancement but overly emphasize the mean brightness. Adaptive histogram equalization (AHE) [20] is based on HE; it processes the image in blocks to address the problem of excessive enhancement. However, due to the independent processing of each pixel block, AHE lacks smooth transitions between blocks, resulting in suboptimal visual effects. CLAHE [21], a generalization of AHE, incorporates a threshold to constrain the contrast, thereby mitigating the problem of noise amplification. In addition, CLAHE utilizes bilinear interpolation to optimize the transitions between blocks, thus resulting in a more harmonious appearance. Although HE-based methods can effectively enhance image contrast, they neglect crucial details, resulting in a diminished portrayal of features such as contours and edge textures within the image.

The frequency domain algorithm is primarily based on multi-scale decomposition (MSD). The methods based on MSD represent various types of spatial and frequency domain information of the source image by decomposing it into different layers. Specific fusion rules are then applied to these different layers to obtain the fused layer. This method is widely applied in the fields of image processing and image quality assessment [22,23,24]. To enhance image details, some state-of-the-art technologies have been proposed. For example, in 2009, Branchitta et al. [25] proposed the bilateral filter and dynamic range partitioning (BF&DRP) algorithm, which utilizes a bilateral filter to decompose raw images into two independent components: the base layer (containing the background) and the detail layer (containing the texture). These two components are processed separately and then combined to reconstruct the final output image. As a result, BF&DRP can retain image details while improving the contrast. However, due to the unstable weight of the bilateral filter kernel near strong edges, gradient reversal artifacts appear in the output image. In 2011, Zuo et al. [26] proposed the bilateral filter and digital detail enhancement (BF&DDE) algorithm, which uses an adaptive Gaussian filter to refine the base and detail layer. However, BF&DDE can only diminish the possibility of gradient reversal artifacts rather than avoiding them completely. In 2014, to achieve detail enhancement, Liu et al. proposed GF&DDE [27], which uses a guided image filter to separate the raw image. However, like BF&DDE, GF&DDE cannot completely eliminate gradient reversal artifacts when the image contains strong edges. In 2016, inspired by the joint bilateral filter, Liu et al. proposed an algorithm called JBF&DDE, which calculates the kernel function by using two adjacent images to distinguish detail information from the raw image. This kernel function is sensitive to the gradient structure within the image, which better enables the elimination of gradient reversal artifacts. In 2020, Xie et al. [28] studied infrared thermal imagers and discovered that their overall response model can be described by a double-exponential statistical fitting model. Consequently, they proposed an algorithm called bi-exponential edge-preserving filtering (BEEPS) to enhance the details of infrared images.

With the rapid development of artificial intelligence, image-processing-method-based CNNs have exhibited outstanding performance. Dong [29] first proposed a convolutional neural network for image super-resolution (SRCNN), which can directly learn an end-to-end mapping between the low- and high-resolution images. Kim [30] found that increasing network depth can significantly improve accuracy and proposed LLCNN based on SRCNN. Zhang [31] developed a skip-connection-based residual channel attention network (RCAN) for image super-resolution, enabling adaptive learning of crucial channel features and enhancing its expressive capabilities. Kuang [32] incorporated a generative adversarial network (GAN) into the conventional CNN framework and introducing IE-CGAN for enhancing single infrared images. This innovative approach effectively mitigates background noise while simultaneously enhancing image contrast and fine details. Wang [33] proposed an innovative target attention deep neural network (TADNN) to achieve discriminative enhancement in an end-to-end manner. However, in practical applications, the calculation of these methods is complex and time consuming. Therefore, the implementation of the aforementioned algorithm is not very hardware friendly.

Due to their parallel computing capabilities, FPGAs have emerged as a promising platform for accelerating computational tasks. Numerous researchers have achieved significant advancements in infrared image enhancement by using FPGAs. For instance, various FPGA-based methods based on CLAHE have been extensively employed to meet real-time processing requirements. Kokufuta et al. [34] processed the image as a whole instead of dividing it into smaller blocks, thereby avoiding interpolation. Unal et al. [35] proposed a look-ahead mechanism for redistribution and redefined the interpolation step to address issues related to image segmentation and correlation interpolation. Chen et al. [36] proposed the use of a fast guided filter and plateau equalization for accelerated enhancement processing. However, this approach introduces gradient reversal artifacts in regions with strong edges. Although the aforementioned methods can achieve fast infrared image enhancement by using FPGAs, the constraints of the enhancement algorithm affect the performance of the enhanced images.

## 2. Proposed Method

### 2.1. Rolling Guidance Filter

Multi-scale image decomposition has been extensively employed in the field of infrared image enhancement. The choice of an appropriate decomposition method considerably affects the performance of the enhanced images. Multi-scale image decomposition involves obtaining images with different levels of blurring through filtering. Commonly used decomposition methods include Gaussian filtering, bilateral filtering [37], guided filtering [38], and WLS filtering [39]. However, these filters do not fully address issues related to noise and gradient reversal. The rolling guidance filter employs a rapidly converging iterative approach to achieve rolling guidance and can produce artifact-free results when separating different scale structures [40]. As a result, it does not rely on local denoising methods but controls the level of detail by controlling the number of iterations. As shown in Figure 1, RGF comprises two main steps: small-structure removal and edge recovery.

#### 2.1.1. Removal of Small Structures

First, the small-structure information in the input image is removed using a Gaussian filter. Assuming that p and q are the pixel coordinates of an image, I is the input image, and G is the output image, the result of Gaussian filtering applied to the input image at the central pixel p can be as follows:(1)$$G(p)=\frac{1}{K_{p}}\sum_{q\in N(p)}\exp(-\frac{{\left\|p-q\right\|}^{2}}{2\sigma_{s}^{2}})I(q),$$
where $K_{p}=\sum_{q\in N(p)}\exp(-\frac{{\left\|p-q\right\|}^{2}}{2\sigma_{s}^{2}})$ is used for normalization, and N(p) is the set of neighboring pixels of p. The structural scale parameter $\sigma_{s}$ is the standard deviation of the Gaussian filter, and the structures whose scale is smaller than $\sigma_{s}$ are removed completely.

#### 2.1.2. Edge Recovery

Edge recovery involves an iterative approach using joint bilateral filtering (JBF) [41] to iteratively recover the blurred large-scale edge structures. During the iterative processing, let $J^{1}=G$. In each iteration, a modified guidance image is obtained from the previous output. All input images are set as I. This iterative processing can be defined as follows:(2)$$J^{t+1}(p)=\frac{1}{K_{p}}\sum_{q\in N(p)}\exp(-\frac{{\left\|p-q\right\|}^{2}}{2\sigma_{s}^{2}}-\frac{{\left\|J^{t}(p)-J^{t}(q)\right\|}^{2}}{2\sigma_{r}^{2}})I(q),$$
where $K_{p}=\sum_{q\in N(p)}\exp(-\frac{{\left\|p-q\right\|}^{2}}{2\sigma_{s}^{2}}-\frac{{\left\|J^{t}(p)-J^{t}(q)\right\|}^{2}}{2\sigma_{r}^{2}})$ is used for normalization, $\sigma_{s}$ and $\sigma_{r}$ respectively control the spatial and range weights, p and q denote the central pixel and neighbor pixels, and $J^{t+1}$ denotes the result of the *t*-th iteration.

In this paper, we defined the RGF operation as follows:(3)$$I_{out}=f_{RGF}(I_{in},\sigma_{s},\sigma_{r},n),$$
where $I_{out}$ is the result of the input image $I_{in}$ after undergoing the RGF operation for n iterations. We found that when the number of iterations n reached a large value, the enhanced image exhibited gradient reversal artifacts. However, setting n=3 resulted in the algorithm achieving the best performance.

### 2.2. Image Enhancement Strategy

The proposed algorithm framework is illustrated in Figure 2. First, the detail layers of the input image are extracted at different scales by using three consecutive rounds of RGF. The output of the third filtering round serves as the base layer. Next, the detail layers from the three scales are merged, and the base layer and detail layers are enhanced. Finally, the enhanced detail layers are combined with the base layer to create an improved infrared image.

#### 2.2.1. Image Decomposition

The smoothing process realized using the RGF can be described as follows:(4)$$I_{B}^{i}=f_{RGF}(I_{in},\sigma_{s}^{i},\sigma_{r}^{i},n),i=1,2,\ldots ,K-1,$$
where $I_{B}^{i}$ represents the result of the i-th filtering. If $\sigma_{s}^{i+1}>\sigma_{s},\sigma_{r}^{i+1}>\sigma_{r}$ then the smoothing degree is $I_{B}^{i+1}>I_{B}^{i}$. As a result, $I_{B}^{i}$ contains more structural information than $I_{B}^{i+1}$. Subsequently, by setting $I_{B}^{0}=I_{in}$, the detailed layers can be extracted using the following relationships:(5)$$I_{D}^{i}=I_{B}^{i}-I_{B}^{i+1},i=1,2,\ldots ,K-1,$$
where $I_{D}^{i}$ represents the detail layers obtained after the *i*-th decomposition, K denotes the decomposition level (in this paper, K=4), and $I_{B}^{K-1}$ is the smoothest version of the original image and serves as base layer $I_{B}$.

#### 2.2.2. Detail Layer Enhancement

(6)$$I_{D_{out}}=\sum_{i=1}^{K-1}I_{D}^{i}\times coe,i=1,2,\ldots ,K-1,$$
where coe is the enhancement coefficient (coe = 3 in this implementation).

#### 2.2.3. Base Layer Enhancement

CLAHE enhances an image by dividing it into multiple sub-blocks and then performing HE on each sub-block. It restricts the degree of contrast enhancement in each sub-block, thereby avoiding excessive enhancement and effectively improving image contrast. CLAHE comprises four main steps: image block division and sub-block histogram statistics, sub-block histogram clipping and redistribution, histogram equalization, and pixel interpolation reconstruction.

For reliable statistical estimation, the size of each sub-block is set as W (W=64 in this implementation) × H (H=64 in this implementation). Next, the histogram of each sub-block is computed using the following formulas:(7)$$h(n)=\sum_{i=0}^{W-1}\sum_{j=0}^{H-1}g(n,i,j),n=0,1,\ldots ,N-1,$$
(8)$$g(n,i,j)=\{\begin{array}{cc} 1 & \mathrm{if}\;I(i,j)=n\\ 0 & \mathrm{otherwise} \end{array},$$
where n is the gray level, histogram bin, (i,j) are the coordinates of a pixel, h(n) is the histogram value for the n-th bin, and g(n,i,j) is the function that determines whether the value of a pixel I(i,j) is equal to n.

A common challenge with standard HE is its tendency to increase the contrast of the sub-regions to the maximum value, resulting in noise amplification. To constrain the contrast of the sub-regions within a certain range and suppress noise, a limiting threshold is introduced, expressed as follows:(9)$$\beta =\frac{M}{N}(1+\frac{\alpha }{100}(S_{\max}-1)),$$
(10)$$S_{\max}=\frac{U}{Q},$$
where β represents the clip limit for each sub-block’s histogram, M represents the number of pixels in each sub-block, N represents the number of gray levels in each sub-block, α is the clip factor, S is a parameter used to control the degree of contrast amplification during the contrast limiting process, U and Q respectively represent the mean and variance of each sub-block. These parameters are used in the calculations to determine the suitable limiting threshold β for each sub-block’s histogram to achieve effective contrast enhancement while suppressing noise. As shown in Figure 3, the portion above β is clipped and redistributed to the bottom of the histogram.

The redistribution algorithm can be represented in the form of pseudocode, as shown in Algorithm 1.


**Algorithm 1:** Redistribution process**Input:** the histogram value *h*(*n*), the clip limit *β***Output:** the histogram value after redistributing *h*(*n*)1. excess = 0;2. **for** (n = 0; n < N; ++ n) {3.  **if** (h[n] > β) {4.   excess += h [n] − β;}}5. m = excess/N;6. **for** (n = 0; n < N; ++ n) {7.  **if** (h[n] < β − m) {8.   h [n] += m;9.   excess −= m;}10.  **else if** (h[n] < β) {11.   excess += h [n] − β;12.   h[n] = β;}}13. **while** (excess > 0) {14.  **for** (n = 0; n < N; ++ n) {15.   **if** (excess > 0) {16.    **if** (h[n] < β) {17.     h [n] += 1;18.     excess −= 1;}}}}excess: the value above the threshold


After performing the contrast limiting process to ensure that the sub-blocks of the histogram do not exceed the clip limit, the cumulative distribution function is computed and pixel value equalization is performed to obtain the new pixel values as follows:(11)$$f_{i,j}(n)=\frac{N-1}{M}\sum_{k=0}^{n-1}h_{i,j}(k),n=1,2,3,\ldots ,N-1,$$
where (i,j) are the coordinates of the sub-block, M is the number of pixels in each sub-block, N is the number of gray levels in each sub-block, and $h_{i,j}(k)$ is the histogram of the image window with coordinates (i,j).

To achieve smoother transitions at block boundaries, interpolation is performed using different methods based on the sub-block’s position. As shown in Figure 4, sub-blocks are categorized into three regions: (1) CR represents sub-blocks that have no connections to others and retain the original pixel mapping function; (2) BR sub-blocks, which undergo linear interpolation for mapping; and (3) IR sub-blocks, which undergo bilinear interpolation based on their four nearest neighboring sub-blocks. The final expression is as follows:(12)$$I_{B_{out}}=\{\begin{array}{c} f_{i,j}(n),\\ LT(f_{i,j}(n)),\\ BT(f_{i,j}(n)), \end{array}\begin{array}{c} n\in CR\\ n\in BR\\ n\in IR \end{array},$$
where $I_{B_{out}}$ represents the result obtained after enhancing the base layer, LT represents linear interpolation, and BT represents bilinear interpolation. The detailed formulas for these two interpolation methods are explained in the Section 4.

#### 2.2.4. Image Reconstruction

To merge the enhanced detail layer and enhanced base layer, the fused image is obtained through inverse transformation as follows:(13)$$I_{EN}=I_{D_{out}}+I_{B_{out}},$$

In summary, the decomposition process is accelerated, the details are enhanced, and the noise is suppressed using the RGF. Finally, the base and the detail components are merged, and an output image with excellent performance is generated.

## 3. Algorithm Experiment and Analysis

To assess the effect and efficiency of the proposed method, we selected a set of infrared images from the TNO Image Fusion Dataset [42] and the M3FD Dataset [43] for experimentation. These selected test datasets comprised diverse scenes, thus offering a comprehensive challenge for the proposed algorithm. We compared the proposed method with five existing infrared image enhancement methods: traditional infrared image enhancement algorithms HE and CLAHE, guided filter-based infrared image enhancement algorithm GF&DDE, the bi-exponential edge-preserving filter-based infrared image enhancement algorithm BEEPS&DDE, and the CNN-based method IE-CGAN. For these methods, we selected the parameters as advised by the authors or through our experience.

### 3.1. Subjective Analysis

Subjective analysis involves assessing the quality of the enhanced image based on an individual’s subjective perception and visual experience. We selected three representative infrared images for a subjective visual evaluation. A high-contrast scene with abundant texture information on rooftops and trees is shown in Figure 5. A scene with strong edges between the person and the surrounding background, which includes mountain peaks with rich textures, is shown in Figure 6. A scene of urban architecture, with tall buildings and towering cranes at great heights, all displaying intricate details, is shown in Figure 7.

The enhancement results obtained using five methods in a high-contrast scene are shown in Figure 5, with the focused areas highlighted by red boxes. In this scene, the HE-based enhancement method improved the contrast of the infrared image but produced overexposure artifacts at the car engine. The CLAHE method effectively enhanced the contrast; however, the details of the houses and trees were not sufficiently prominent. The GF&DDE method performed well in smoothing background noise and enhancing contrast; however, the presence of gain masks caused the smoothing out of some details in the regions of interest. The IE-CGAN method performs well in image denoising but loses some information of the fine details. Compared with the other four enhancement methods, the images processed using the BEEPS&DDE method and the proposed method exhibited rich texture details, such as the abundant leaf details on trees. However, in terms of overall image performance, the proposed method exhibited higher contrast and better representation.

The enhancement effects of the enhancement algorithms on the “thermal” image are shown in Figure 6. The HE algorithm yielded higher overall contrast among all the enhancement algorithms; however, it produced overexposure artifacts on the thermal target, resulting in a considerable loss of fine-grained details in the target (e.g., the person enclosed within the red box lacks discernible details). Although the CLAHE algorithm effectively mitigated overexposure caused by HE and yielded relatively favorable results in terms of contrast enhancement, it struggled in preserving intricate details, consequently resulting in a somewhat blurred perception. The IE-CGAN method enhances the contrast of image but the visual improvement is not very pronounced. In contrast, GF&DDE and BEEPS&DDE effectively improved the overall brightness. GF&DDE slightly outperformed BEEPS&DDE in handling thermal targets, whereas the latter excelled in enhancing texture information, such as shrubs and mountains in the background. The proposed algorithm greatly improved image contrast and exhibited a better effect on detail enhancement and maintenance (e.g., the details of the mountain peaks and the person in the image). Furthermore, the outline of the infrared target was visible without gradient reversal artifacts, thereby demonstrating its excellent visual effect.

The enhancement results obtained using the enhancement algorithms on urban scenes are shown in Figure 7. Image processing using HE yielded a very bright image, and a lot of the detailed information about the target scene was lost. CLAHE yielded visually pleasing results but failed to improve the perceptibility of the small details in the image. GF&DDE performed well in noise suppression, whereas BEEPS&DDE excelled in highlighting texture details. Although both algorithms improved the overall brightness to a certain extent, the overall contrast of the image was not high, and the detailed information was not sufficiently prominent, such as the details of the bushes in the lower right corner. In this scenario, the performance of the IE-CGAN method was not satisfactory, which may be attributed to the insufficiency of the training dataset. The proposed algorithm improved the contrast and clarity of different areas of the image to different degrees, such as the edge outline of the tower crane being more explicit and the contrast of the building part being improved. The proposed algorithm yielded an image wherein the details of the scene were highlighted and the visual effect was more realistic.

To validate the applicability of the proposed enhancement algorithm across various scenarios, we performed a comparative analysis by evaluating six methods in seven different scenes, such as texture-rich wire fences, streets with numerous thermal targets, and dense forests with intricate texture details. As can be seen from the enhancement results of these methods applied to the seven scenes shown in Figure 8, the proposed method outperformed the other five methods in terms of enhancement performance.

### 3.2. Objective Analysis

Currently, the field of image processing has become a research hotspot, and assessing the quality of processed images remains a challenge. Quality image assessment (IQA) methods can be categorized into subjective and objective ones [44]. Since the fact that the human visual system is the ultimate recipient of visual signals, subjective evaluation is usually the most accurate and reliable method. However, because subjective test consumes significant resources, it is typically not employed as an optimization metric in practice. Objective quality assessment methods are usually designed or trained using subjective evaluation data. They serve as an ideal approach for timely image performance assessment and optimizing. Objective quality assessment can be divided into traditional metrics such as PSNR, SSIM, MSE, and so on, and emerging metrics such as UCA [45], BPRI [46], BMPRI [47], and so on.

To objectively evaluate the enhancement effects of the different methods in the ten aforementioned infrared scenes, five traditional image evaluation metrics were employed, such as average gradient (AG) [48] and edge intensity (EI) [49], which are based on image features; figure definition (FD), which quantifies the level of detail and distinctness present in the visual content of the image; and root mean square contrast (RMSC) [50], which quantifies the contrast level of the image. These metrics are widely used for evaluating the quality of an image. The evaluation results are presented in Table 1, Table 2, Table 3 and Table 4. The average values of all evaluation parameters are presented in Table 5, and the optimal value of each parameter is marked in bold.

AG represents the average magnitude of variations in pixel values across the image. A higher AG value indicates that the enhancement effect of this algorithm contains richer gradient information and detailed textures. The AG calculation results are presented in Table 1. The formula for calculating AG is as follows:(14)$$AG=\frac{\sqrt{2}}{2(M-1)(N-1)}\sum_{i=1}^{M-1}\sum_{j=1}^{N-1}\sqrt{{\left(\frac{\partial I(i,j)}{\partial i}\right)}^{2}+{\left(\frac{\partial I(i,j)}{\partial j}\right)}^{2}},$$
where (i,j) is a coordinate of the image, and $\frac{\partial I(i,j)}{\partial i}$ and $\frac{\partial I(i,j)}{\partial j}$ are the horizontal and vertical gradient values, respectively. M and N are the height and width of the image, respectively.

EI refers to the strength or magnitude of the edges in the image. A higher EI value indicates that the image has higher contrast and more abundant detail information. The calculation EI results are listed in Table 2. The formula of EI is as follows:(15)$$EI=\frac{1}{MN}\sqrt{\sum_{i=1}^{M}\sum_{j=1}^{N}\left(s_{x}{\left(i,j\right)}^{2}+s_{y}{\left(i,j\right)}^{2}\right)},$$
where $s_{x}(i,j)$ and $s_{y}(i,j)$ are Sobel operators for the *x* and *y* directions, respectively.

FD quantifies the level of detail and distinctness in the image. A higher FD value indicates that the image contains high levels of sharpness and visual information. The FD calculation results are presented in Table 3, and the formula for calculating FD is as follows:(16)$$FD=\frac{1}{MN}\sum_{i=1}^{M-1}\sum_{j=1}^{N-1}\sqrt{\frac{{\left(I(i+1,j)-I(i,j)\right)}^{2}+{\left(I(i,j+1)-I(i,j)\right)}^{2}}{2}}$$

RMSC is used to evaluate the degree of image denoising and enhancement. The larger the value of RMSC, the higher the contrast of the image. The proposed algorithm yielded a high RMSC value, thus indicating that it effectively increases the contrast of infrared images. The RMSC calculation results are presented in Table 4. The formula for calculating RMSC is as follows:(17)$$RMSC=\sqrt{\frac{1}{MN}\sum_{i=1}^{M}\sum_{j=1}^{N}{\left(I(i,j)-\bar{I}(i,j)\right)}^{2}},$$
where $\bar{I}$ is the average intensity of all pixel values of the experiment image.

The average values of the four aforementioned metrics obtained by applying six different methods to enhance ten infrared images are presented in Table 5. These metrics were used to objectively evaluate the performance of each method. As can be observed from the values in Table 5, the proposed method outperformed the others in terms of AG, EI, and FD values, thus indicating its superiority in enhancing image texture details and improving image clarity. However, the enhanced images generated by the proposed method did not have a high RMSC value when compared to other methods. This can be attributed to the adoption of the CLAHE method in the base layer, which effectively maintains the overall contrast within an appropriate range. In contrast, the other two decomposition-based image enhancement methods use the HE method at the base layer, resulting in higher overall contrast but sometimes causing overexposure in certain images. This overexposure results in a relatively poor overall visual perception of the enhanced images. Therefore, it is evident that the proposed method possesses distinct advantages in terms of increasing the contrast and enhancing the edge details compared to the other methods.

## 4. Hardware Implementation

### 4.1. Hardware Architecture

To facilitate swift algorithm functionality validation and optimization, we designed and implemented the image enhancement module by using the high-level synthesis (HLS) tool, which can convert high-level programming languages (C/C++) into hardware description languages (HDL/VHDL), thereby elevating the level of abstraction and offering advantages such as shorter development cycles, increased development efficiency, and simplified algorithm hardware implementation.

The hardware architecture of the proposed method is shown in Figure 9. For hardware implementation, we used AXU15EG as the development platform. The heterogeneous architecture includes a processing system (PS) and programmable logic (PL). In the PS, an ARM processor performs system control and scheduling tasks, such as data preprocessing, IP configuration, and image streaming. The PL includes the RGF module and the CLAHE module, which are used for enhancing infrared images. The AXI bus facilitates high-speed communication and data interaction between PS and PL components. Video direct memory access is used for reading infrared images and storing enhanced images. To achieve computational optimization, dataflow instructions are used to optimize the processing flow. These instructions ensure that the intermediate data generated in each processing stage are stored using FIFO buffers. This approach enables parallel processing between the modules, thereby facilitating efficient data handling and promoting parallelization among the processing stages.

### 4.2. RGF Unit Design

The RGF process involves two main steps. First, Gaussian filtering is employed to remove small structures, followed by joint bilateral filtering to restore edges. To achieve a balance between resource allocation and filtering performance, we selected a 5 × 5 filter kernel. The architecture of the RGF is shown in Figure 10. The input pixel data are cached through row buffers. Four row buffers are required to accommodate the 5 × 5 filter kernel, and the data in these buffers are used for calculations within the processing window.

In the first step of RGF, the row calculation unit requires only the original pixel values as the input. The result of Gaussian filtering is then calculated using the 5 × 5 Gaussian filter. Subsequently, in the joint bilateral filtering process, the row calculation unit takes the original pixel values of the input image and the pixel values of the previously computed guidance image. Unlike Gaussian filtering, joint bilateral filtering considers both spatial and grayscale weights, enabling the removal of small structures while restoring large-scale edge information. The five row calculation units produce the results for the current window, which are then sent to Sum2 for accumulation. The normalization coefficient results are sent to Sum1 for accumulation. Finally, the calculation results are divided by the normalization coefficient to obtain the filtered output pixel value. This iterative process is continued until the entire restoration process is completed.

The spatial weight and range kernel of the guidance image are denoted as $W_{s}$ and $W_{r}$, respectively, and their formulas are as follows:(18)$$\{\begin{array}{c} W_{s}=\exp(-\frac{{\left\|\left(i-m)+(j-n\right)\right\|}^{2}}{2\sigma_{s}^{2}})\\ W_{r}=\exp(-\frac{{\left\|f_{guide}(i,j)-f_{guide}(m,n)\right\|}^{2}}{2\sigma_{r}^{2}}) \end{array},$$
where (m,n) represents the pixel coordinates within a 5 × 5 neighborhood, (i,j) represents the coordinates of the center pixel, and $f_{guide}$ represents the guide image.

As can be observed from Equation (18), division and exponentiation operations are required to compute the spatial and range kernel within the processing window. To reduce computational load, the precomputed results can be stored in a ROM, enabling the calculation results to be obtained through LUTs. The row calculation unit design is shown in Figure 11. To ensure high processing speed, we implemented parallel computations for all five row calculation units and their five corresponding cached pixels.

For the LUT implementation of the spatial and range kernel, we set $\sigma_{s}=40,\sigma_{r}=0.1$. To facilitate computation, we scaled the obtained floating-point results by a factor of 256 and then right-shifted the final output result by eight bits to obtain the desired result. Because the values of $f_{guide}(i,j)-f_{guide}(m,n)$ lie within the range of [0, 255], we were able to directly determine the results of $W_{r}$ and stored them in a ROM. As can be seen from Figure 12, when the pixel value differences exceeded a certain threshold (in this paper, 86), the corresponding output results tended toward 0. Leveraging this characteristic, we optimized the LUTs by setting the output to 0 for pixel value differences exceeding 86, thereby reducing the amount of data stored in the table by approximately 66%. This optimization greatly minimized the hardware resources required for our approach.

### 4.3. CLAHE Unit Design

To meet real-time requirements, the CLAHE algorithm has been designed with a focus on parallel computation and pipelining. The modules are interconnected using hls::stream, which enables data flow between them. By incorporating dataflow directives, the HLS tool synthesizes the design to enable overlapping execution, thereby maximizing the utilization of available resources and improving the overall throughput.

#### 4.3.1. Histogram Calculation

First, the input image is partitioned into sub-block regions, as shown in Figure 13.

In our implementation, the resolution of the infrared images is 640 × 480 pixels. The input image is divided into 12 sub-blocks, each measuring 160 × 160 pixels. Subsequently, histogram statistics are computed for each sub-block. The obtained results are then inputted to the sub-block histogram clipping and redistribution module.

#### 4.3.2. Histogram Clipping and Redistribution

The sub-block histogram clipping and redistribution module is illustrated in Figure 14. The caching and histogram statistics of each sub-block are computed before being fed into the histogram clipping unit, which then calculates the total sum excess of pixel values in the range of 0–255. This excess sum is evenly redistributed across the intervals, and the results are stored in a dual-port RAM. This iterative process is continued until the values within each interval no longer exceed the clipping threshold, indicating the completion of the computation. To enhance the processing speed, parallel execution is employed for the sub-blocks, with a dedicated dual-port data cache unit allocated for each one.

#### 4.3.3. Mapping Function

To enhance the efficiency and optimize on-chip memory usage, a row-based buffering strategy is employed instead of a frame-based approach for the hardware implementation of the CLAHE algorithm. This design addresses the problem of uneven enhancements between adjacent image blocks by introducing interpolation between them.

Bilinear interpolation is employed in the interpolation circuit for most sub-blocks, necessitating the caching of mapping functions from the four surrounding sub-blocks. To achieve this, mapping functions of at least two rows of sub-blocks are stored in buffers. In addition, a dedicated buffer is used to seamlessly receive mapping functions for the subsequent sub-block.

As shown in Figure 15, the pipeline incorporates three buffers to enable continuous interpolation operations and thereby enhance the system’s operating frequency. The caching procedure follows a three-cycle pattern: Cycle N, Cycle N + 1, and Cycle N + 2. In Cycle N, Line Buffers N and N + 1 store two rows of sub-blocks required for interpolation, and Line Buffer N + 2 caches the mapping functions of the next sub-block. In Cycle N + 1, interpolation calculations are performed for Line Buffers N + 1 and N + 2, and Line Buffer N is cleared to accommodate the data of the next row of sub-blocks. In Cycle N + 2, Line Buffer N + 2 is cleared for caching Line Buffer N + 4 data, and interpolation results for Line Buffers N + 2 and N + 3 are computed. This three-cycle loop is continued until the interpolation process covers the entire image and the final enhanced result is obtained. This approach makes the interpolation operation highly efficient, resulting in improved system performance in terms of operating frequency and optimal utilization of on-chip storage resources in hardware implementations of the CLAHE algorithm.

#### 4.3.4. Interpolation

The pixel interpolation reconstruction module involves two steps. First, the weights are calculated. Next, the interpolation calculations are performed. As shown in Figure 16, different interpolation methods are employed based on the sub-block’s position.

For sub-blocks situated in the corners of the image (CR), interpolation is performed using the sub-block’s mapping function. For sub-blocks situated along the image edges (BR), linear interpolation is performed using the mapping functions of the two surrounding sub-blocks. For the majority of sub-blocks (IR), bilinear interpolation is performed.

First, interpolation is performed in the *x*-direction by using the following formula:(19)$$\{\begin{array}{c} f(R_{1})=\frac{x_{2}-x}{x_{2}-x_{1}}f(Q_{11})+\frac{x-x_{1}}{x_{2}-x_{1}}f(Q_{21})\\ f(R_{2})=\frac{x_{2}-x}{x_{2}-x_{1}}f(Q_{12})+\frac{x-x_{1}}{x_{2}-x_{1}}f(Q_{22}) \end{array},$$
where $R_{1}=(x,y_{1})$ and $R_{2}=(x,y_{2})$.

Next, interpolation is performed in the *y*-direction by using the following formula:(20)$$f(P)=\frac{y_{2}-y}{y_{2}-y_{1}}f(R_{1})+\frac{y-y_{1}}{y_{2}-y_{1}}f(R_{2}),$$

Finally, the interpolation result is obtained using the following formula:(21)$$\begin{array}{c} f(x,y)=\frac{f(Q_{11})}{\left(x_{2}-x_{1})(y_{2}-y_{1}\right)}(x_{2}-x)(y_{2}-y)+\frac{f(Q_{21})}{\left(x_{2}-x_{1})(y_{2}-y_{1}\right)}(x-x_{1})(y_{2}-y)\\ +\frac{f(Q_{12})}{\left(x_{2}-x_{1})(y_{2}-y_{1}\right)}(x_{2}-x)(y-y_{1})+\frac{f(Q_{22})}{\left(x_{2}-x_{1})(y_{2}-y_{1}\right)}(x-x_{1})(y-y_{1}) \end{array}$$

In the absence of optimization, the interpolation process requires a considerable number of multiplier resources. To enhance the efficiency of the interpolation process and make it more suitable for implementation, the bilinear interpolation formula must be revised. Let the weights in the horizontal and vertical directions be denoted as α and β, respectively:(22)$$\{\begin{array}{l} \alpha =\frac{y-y_{1}}{y_{2}-y_{1}}\\ \beta =\frac{x_{2}-x}{x_{2}-x_{1}} \end{array}$$

Equation (21) can be transformed as follows:(23)$$\begin{array}{ll} f(x,y) & =\alpha (\beta f(Q_{12})+(1-\beta )f(Q_{22})+(1-\alpha )(\beta f(Q_{11})+(1-\beta )f(Q_{21}))\\ & =\alpha \beta f(Q_{12})-\alpha \beta f(Q_{22})+\alpha f(Q_{22})-\alpha \beta f(Q_{11})\\ & +\alpha \beta f(Q_{21})-\alpha f(Q_{21})+\beta f(Q_{11})-\beta f(Q_{21})+f(Q_{21})(Q_{21})\\ & =\alpha (\beta (f(Q_{12})-f(Q_{22}))+f(Q_{22}))-\alpha (\beta (f(Q_{11})-f(Q_{21}))+f(Q_{21}))\\ & +\beta (f(Q_{11})-f(Q_{21}))+f(Q_{21}) \end{array}$$
(24)$$\{\begin{array}{l} P_{1}=\beta (f(Q_{12})-f(Q_{22}))+f(Q_{22})\\ P_{2}=\beta (f(Q_{11})-f(Q_{21}))+f(Q_{21}) \end{array}$$

Substituting Equation (23) into Equation (24), the final formula can be simplified as follows:(25)$$f(P)=\alpha (P_{1}-P_{2})+P_{2}$$

The optimized interpolation unit is illustrated in Figure 17. After optimization, the interpolation unit requires only three multipliers, three subtractors, and three adders. Variables $f(Q_{11})$ and $f(Q_{12})$ are obtained from the cached mapping function values of the previous row in the Line Buffer, whereas variables $f(Q_{21})$ and $f(Q_{22})$ are obtained from the current row in the Line Buffer. After the completion of computation for each row of sub-blocks, the Line Buffer is updated according to the pattern shown in Figure 15, finalizing the computations for the entire image. The “Weights” component in Figure 15 is a division unit that is used to generate the weights for rows and columns according to the input pixel address, representing the parameters α and β, respectively, in Equation (25).

### 4.4. FPGA Implementation Results

The proposed algorithm is implemented on the AXU15EG development board with AMD Xilinx Zynq UltraScale+ XCZU15EG-FFVB1156–2-I MPSoC device. Throughout the design process, the utmost care is taken to preserve data accuracy to prevent any significant loss in data precision and ensure that the integrity of the enhanced images remains intact. Resource utilization details of the developed image enhancement module implemented on the FPGA are presented in Table 6.

From Table 6, it can be observed that the utilization percentages of BRAM_18K and LUT are relatively higher compared to DSP48E and FF resources. This is attributed to the consideration of real-time applications during the architecture design process. To enhance processing speed, on-chip caching of image data was implemented, resulting in a higher utilization of BRAM_18K resources. As illustrated in Figure 12, weight data was preloaded into the LUT, thereby eliminating a portion of nonlinear operations, leading to higher LUT resource utilization while reducing the utilization of DSP48E resources. Additionally, we simplified the bilinear interpolation algorithm, which can reduce the utilization of DSP48E and FF resources.

As can be observed from the processing speeds achieved using FPGA and PC platforms (Table 7), the image enhancement module exhibited a processing speed of approximately 6.86 ms (147 fps) when operating under a 114 MHz reference clock. In comparison to the processing speed achieved on a PC, the FPGA-based processing speed was approximately 29.4 times faster, thereby enabling nearly real-time output of the enhanced image.

The enhanced infrared images of three scenes on the PC and FPGA platforms are shown in Figure 18. Overall, the enhanced images obtained from both platforms exhibited good visual representation. However, due to hardware limitations, there were some differences in the results. Compared to the enhancement results on the PC, the FPGA-enhanced images exhibited poorer contrast and detail processing. For instance, in the first scene, the house appeared darker, and the targets within the red boxes in the second and third scenes appeared blurry. Despite a minor precision loss in the FPGA enhancement results, the overall visual representation and enhancement speed of the processed images were within acceptable ranges.

The average metrics of the enhanced images obtained using different platforms in the three aforementioned test datasets are presented in Table 8. FPGA’s enhancement results were inferior to those of PC in terms of all four metrics: AG, EI, FD, and RMSC. The objective analysis results were consistent with the subjective analysis results, thus indicating that FPGA’s enhancement results suffer only minor losses in texture and detail information along with a decrease in contrast, thereby resulting in an overall performance reduction in the enhanced images. From the results, it can be concluded that FPGA achieves a good balance between enhancement effectiveness, resource consumption, and enhancement speed.

## 5. Discussion

In the field of infrared image enhancement, enhancement quality and speed are of the utmost importance. Many advanced algorithms have been proposed for improving the performance of infrared images. However, their high computational complexity results in decreased enhancement speed. Thus, achieving a balance between image enhancement quality and speed to meet real-time application requirements remains a challenge. Our research has a very broad range of applications, such as military security, medical diagnostics, and autonomous driving, making it highly meaningful. Furthermore, in future research, we can apply this technology to multimodal image fusion techniques and other areas within the field of image processing.

## 6. Conclusions

In this paper, we proposed a novel method for infrared image enhancement and implemented it on an FPGA. Compared with other enhancement methods, the proposed method exhibits superior performance in enhancing details, improving contrast, and reducing gradient reversal artifacts. In the proposed method, first, the image is decomposed into a base layer and multiple detail layers of different scales by using the RGF. Detail enhancement factors are used for the detail layers, whereas CLAHE is used for the base layer. Finally, the enhanced images from each layer are fused, yielding an image with globally enhanced details from the input image. For deploying the proposed algorithm on an FPGA, we adopted a parallel dataflow approach for image processing and strived to minimize the utilization of hardware resources. The proposed method yielded enhanced images with excellent expressiveness, with each image having a resolution of 640 × 480 pixels, achieving a processing speed of 147 fps. Due to its real-time processing capability, the proposed method offers a feasible solution for real-time scenarios.

## Figures and Tables

**Figure 1 sensors-23-08101-f001:**
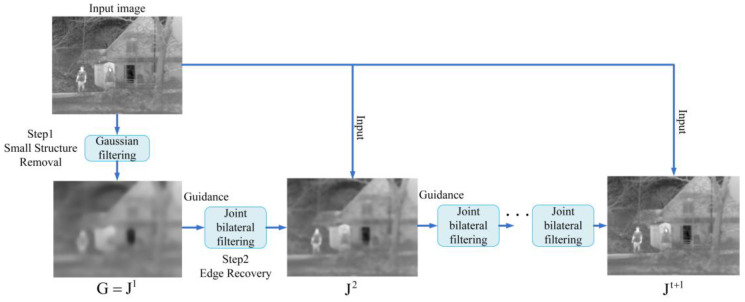
Rolling guidance filter.

**Figure 2 sensors-23-08101-f002:**
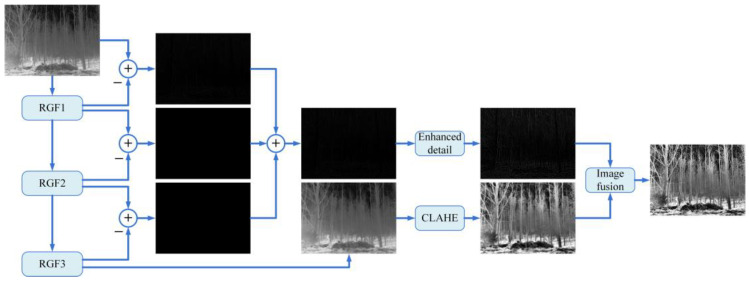
The proposed framework.

**Figure 3 sensors-23-08101-f003:**
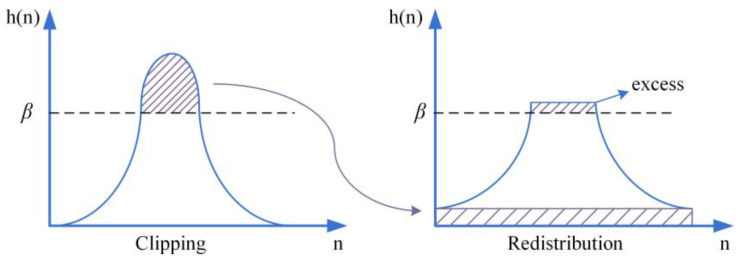
Histogram clipping and redistribution.

**Figure 4 sensors-23-08101-f004:**
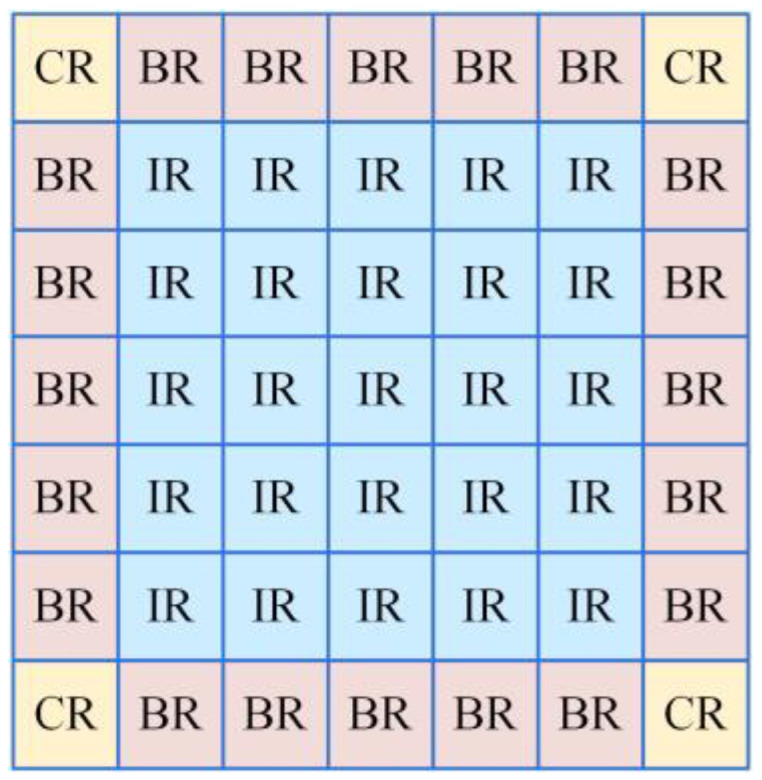
Division of sub-blocks.

**Figure 5 sensors-23-08101-f005:**
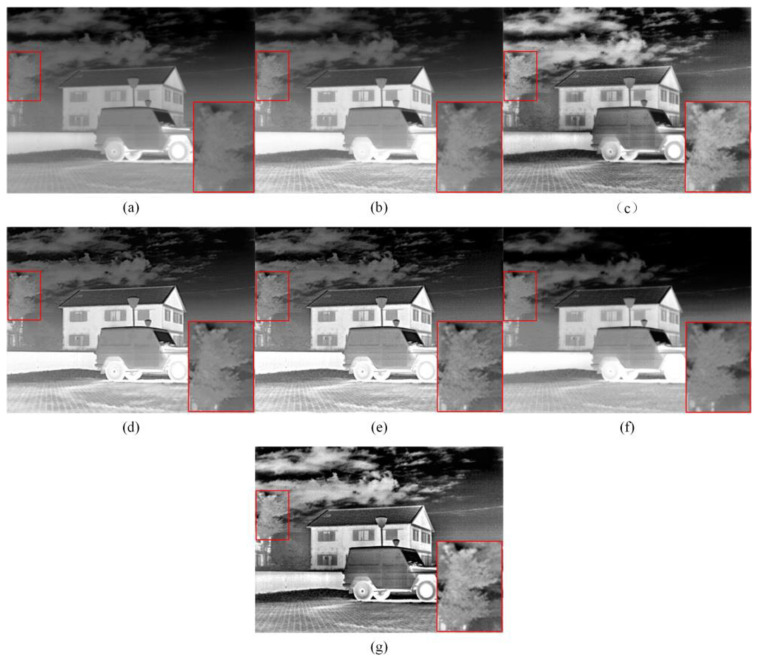
Enhanced images in the first scene using five different methods. Within the red frame is a tree with rich texture details. (**a**) Original image; (**b**) HE; (**c**) CLAHE; (**d**) GF&DDE; (**e**) BEEPS&DDE; (**f**) IE-CGAN; (**g**) Proposed.

**Figure 6 sensors-23-08101-f006:**
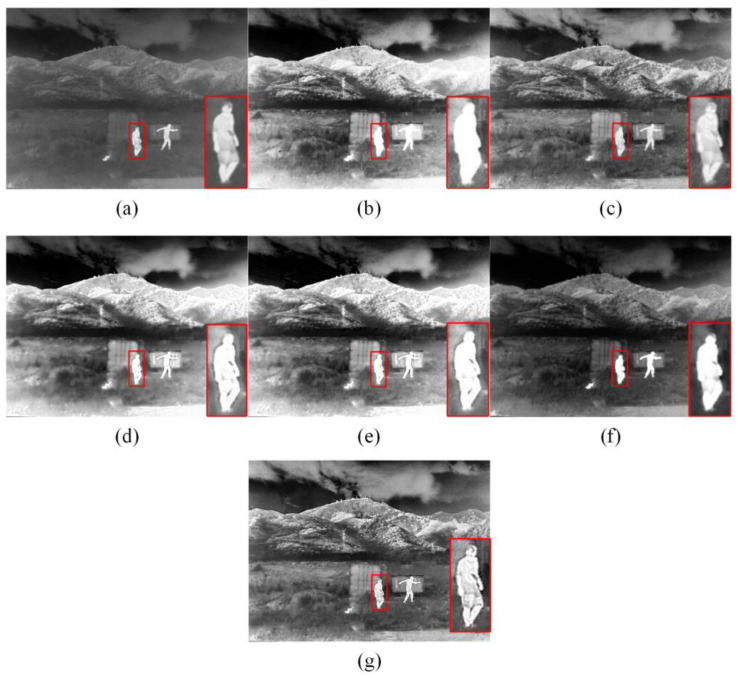
Enhanced images in the second scene using five different methods. Within the red frame is the thermal target. (**a**) Original image; (**b**) HE; (**c**) CLAHE; (**d**) GF&DDE; (**e**) BEEPS&DDE; (**f**) IE-CGAN; (**g**) Proposed.

**Figure 7 sensors-23-08101-f007:**
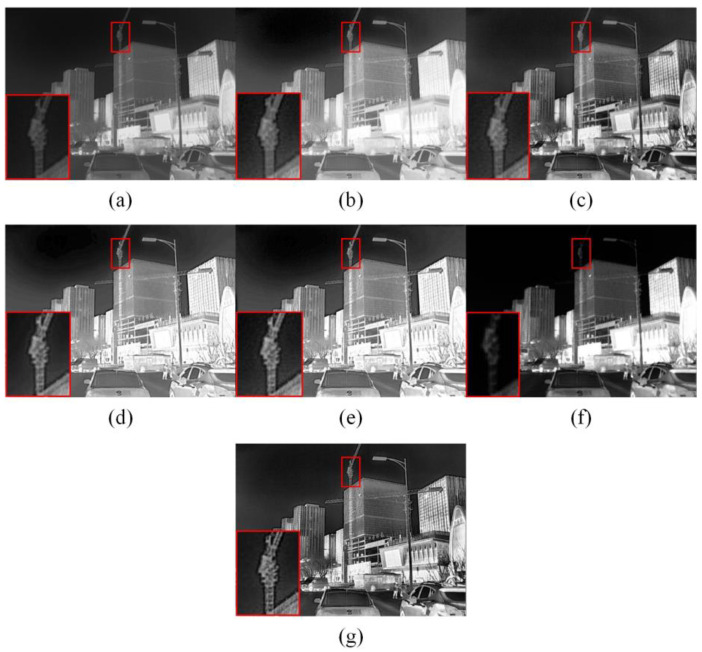
Enhanced images in the third scene using five different methods. The red frame shows the tower cranes that we are more interested in. (**a**) Original image; (**b**) HE; (**c**) CLAHE; (**d**) GF&DDE; (**e**) BEEPS&DDE; (**f**) IE-CGAN; (**g**) Proposed.

**Figure 8 sensors-23-08101-f008:**
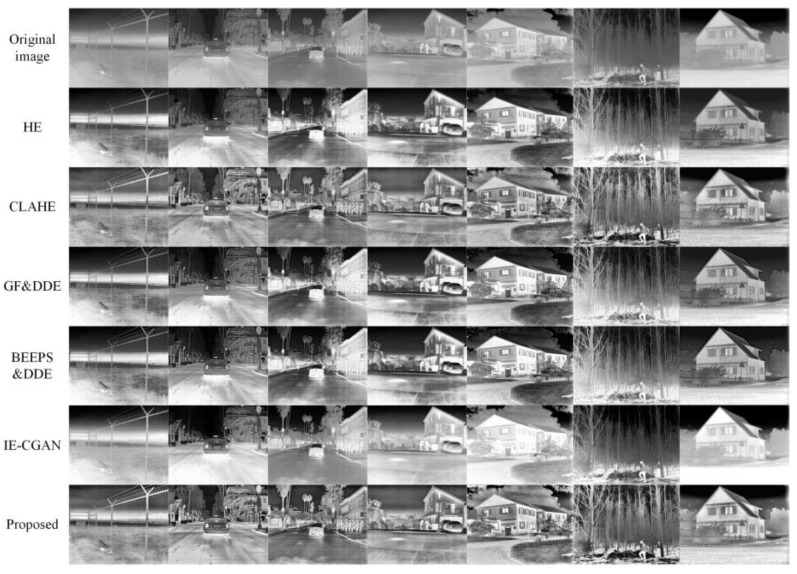
Enhanced images in the other seven scenes using six different methods.

**Figure 9 sensors-23-08101-f009:**
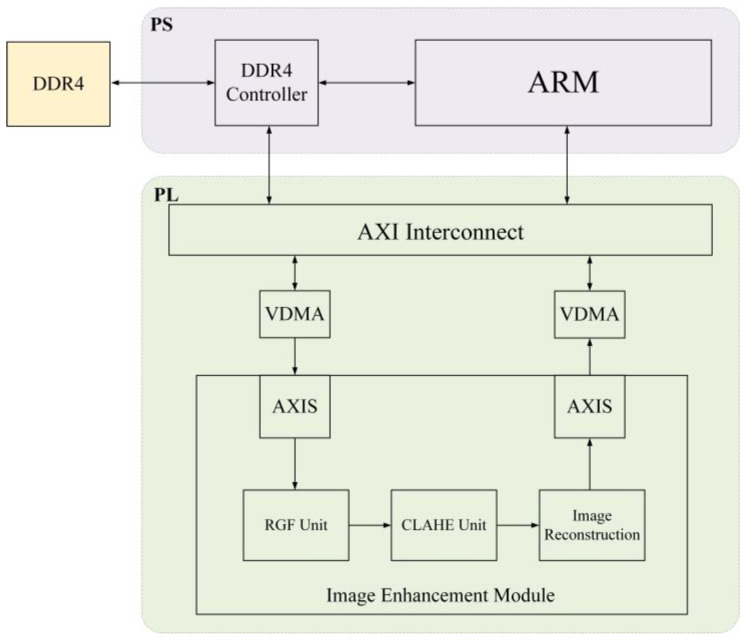
Hardware architecture.

**Figure 10 sensors-23-08101-f010:**
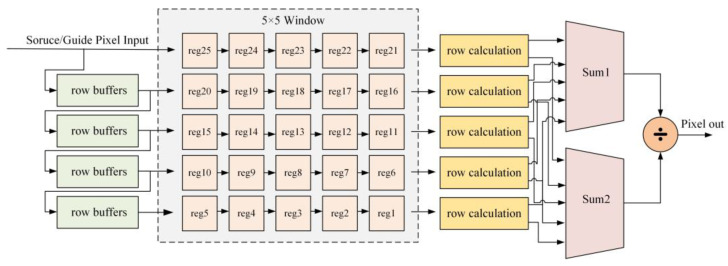
Schematic diagram of RGF unit.

**Figure 11 sensors-23-08101-f011:**
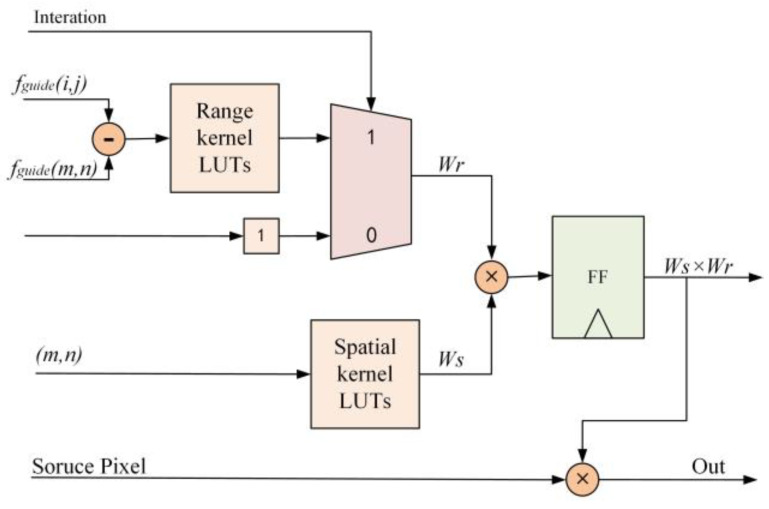
Schematic diagram of row calculation unit.

**Figure 12 sensors-23-08101-f012:**
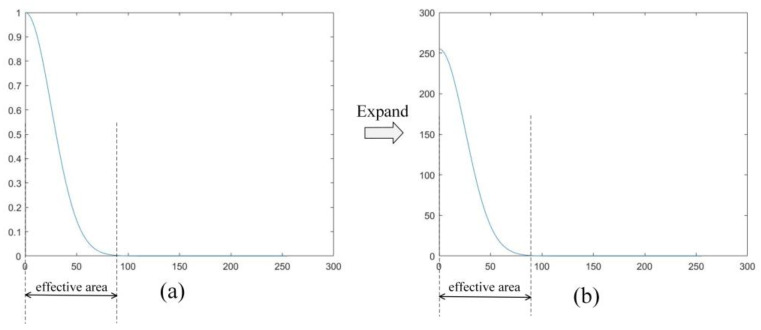
Optimization of LUTs. (**a**) Curve before expand the difference. (**b**) Curve after expand the difference.

**Figure 13 sensors-23-08101-f013:**
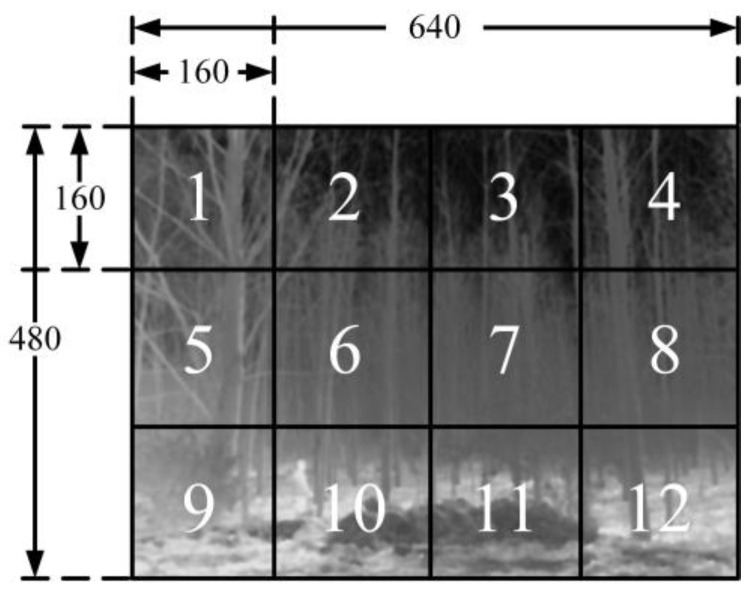
Schematic diagram of sub-block division.

**Figure 14 sensors-23-08101-f014:**
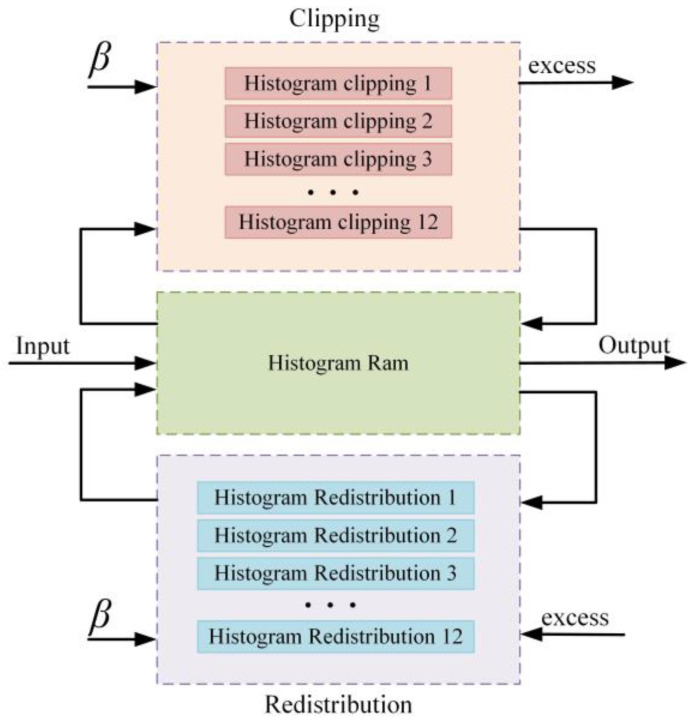
Schematic diagram of sub-block clipping and redistribution.

**Figure 15 sensors-23-08101-f015:**
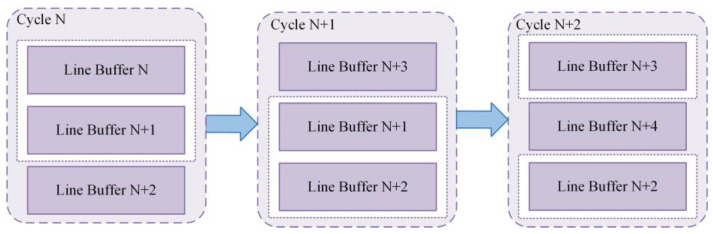
Mapping function RAM module.

**Figure 16 sensors-23-08101-f016:**
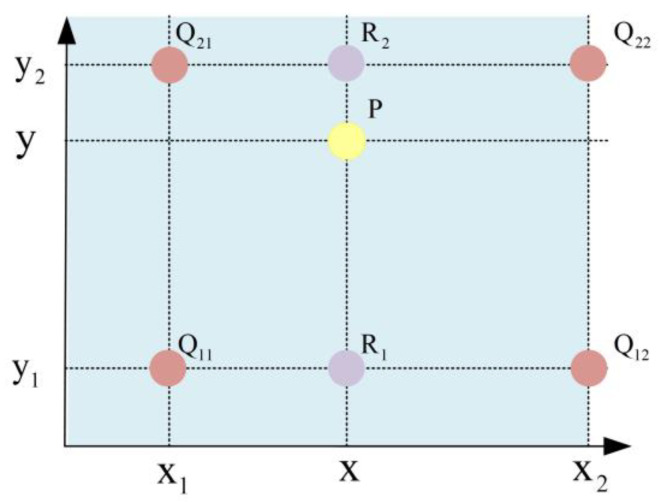
Schematic diagram of bilinear interpolation.

**Figure 17 sensors-23-08101-f017:**
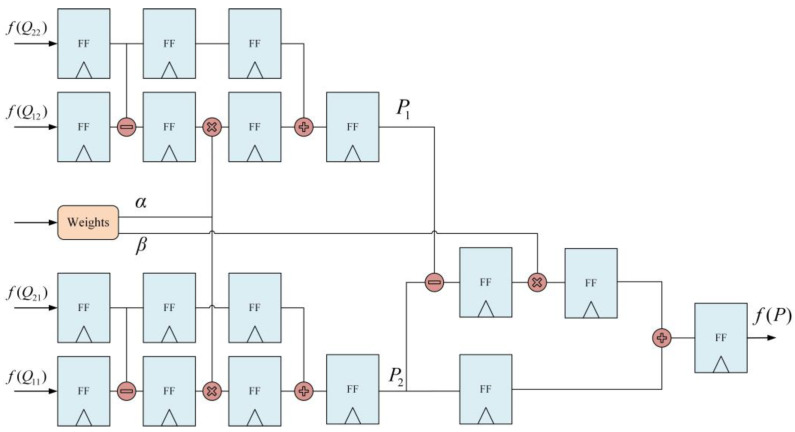
Pixel interpolation reconstruction unit.

**Figure 18 sensors-23-08101-f018:**
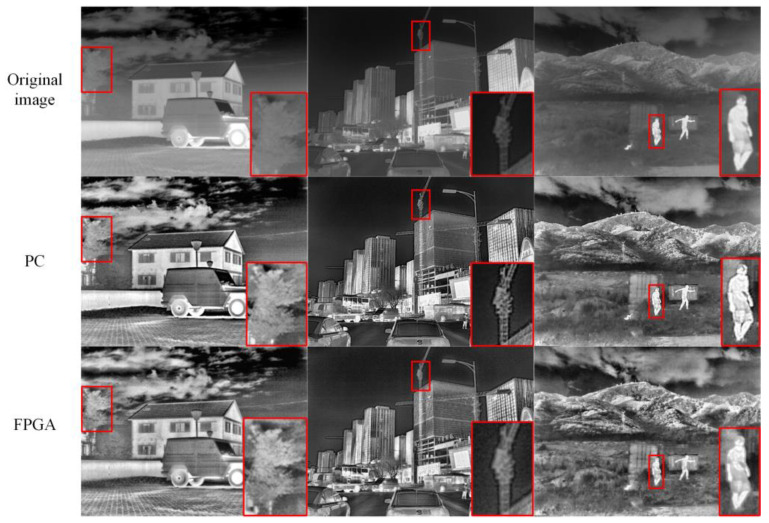
Test results of PC and FPGA. The red frame represents the area of interest.

**Table 1 sensors-23-08101-t001:** The AG values of images.

	HE	CLAHE	GF&DDE	BEEPS&DDE	IE-CGAN	Proposed
Scene1	3.3064	5.75	4.7207	6.162	3.0171	9.4747
Scene2	4.7369	7.7667	8.6033	9.9665	2.3355	10.6338
Scene3	5.1166	5.515	4.7766	6.1097	4.8409	6.807
Scene4	6.591	9.8609	9.1665	11.1179	3.1194	11.2412
Scene5	4.6784	8.901	7.4432	9.2324	5.2663	9.6896
Scene6	6.7287	8.808	8.0154	9.7518	4.1704	10.0586
Scene7	6.7745	7.5473	6.5898	8.4726	4.6242	8.1753
Scene8	6.2241	8.6474	7.5217	9.2147	3.8121	9.4747
Scene9	8.0002	13.6018	12.2886	13.8019	4.071	14.7438
Scene10	3.9875	5.8685	4.7549	6.7469	6.1466	6.351

**Table 2 sensors-23-08101-t002:** The EI values of images.

	HE	CLAHE	GF&DDE	BEEPS&DDE	IE-CGAN	Proposed
Scene1	34.7727	60.3802	49.837	63.9985	32.1226	97.3956
Scene2	48.8454	81.3168	91.5198	104.4826	24.3485	112.2565
Scene3	51.934	55.893	49.5355	62.0235	52.3518	67.612
Scene4	62.5707	92.6684	86.2605	103.9665	32.9632	105.8907
Scene5	45.44	86.6194	72.4972	88.771	52.4775	94.9481
Scene6	65.1722	85.4135	79.2866	95.0336	43.7175	98.7767
Scene7	66.2712	72.4612	64.925	82.1765	47.7164	79.061
Scene8	63.9582	88.6129	77.6044	93.9924	39.005	97.3956
Scene9	84.0962	143.5655	129.0542	144.5164	42.2968	155.592
Scene10	40.2207	59.1701	48.294	67.3686	65.662	64.0221

**Table 3 sensors-23-08101-t003:** The FD values of images.

	HE	CLAHE	GF&DDE	BEEPS&DDE	IE-CGAN	Proposed
Scene1	3.7445	6.4858	5.3914	7.0231	3.5633	11.0912
Scene2	5.8501	9.3404	10.0945	11.9584	2.8753	12.5943
Scene3	6.3087	6.7871	5.6667	7.4641	5.5845	8.7062
Scene4	9.4174	14.0102	12.7917	15.6909	3.6518	15.8017
Scene5	6.7108	12.6621	10.5362	13.1964	6.7881	13.6354
Scene6	9.5789	12.4054	10.907	13.5917	5.373	13.8656
Scene7	9.6476	10.888	9.0738	11.8983	5.939	11.6588
Scene8	7.3154	10.1447	8.8202	10.8575	5.0632	11.0912
Scene9	9.1861	15.4576	14.0613	15.8	4.927	16.7094
Scene10	4.8271	7.0696	5.7111	7.9857	7.0285	7.65

**Table 4 sensors-23-08101-t004:** The RMSC values of images.

	HE	CLAHE	GF&DDE	BEEPS&DDE	IE-CGAN	Proposed
Scene1	74.5077	58.695	78.034	82.9624	83.5156	58.9593
Scene2	74.9025	62.5509	81.5486	85.7531	95.0626	75.8319
Scene3	74.6886	46.0226	77.7295	80.7774	82.7147	50.6077
Scene4	74.9016	49.5487	79.2614	83.9679	49.5168	58.9766
Scene5	74.8368	54.0383	78.8685	83.7585	75.446	62.5467
Scene6	74.6253	46.9971	79.7722	83.2655	90.9194	57.3257
Scene7	74.7973	46.8248	78.2011	83.1262	63.522	53.9331
Scene8	74.6965	51.6546	79.214	83.8128	70.1383	58.9593
Scene9	74.787	56.3033	81.8486	85.9174	60.9261	65.8268
Scene10	72.1822	68.7966	75.1887	84.2537	71.5311	74.834

**Table 5 sensors-23-08101-t005:** Objective analysis for different methods.

Algorithm	AG	EI	FD	RMSC
HE	5.6144	56.3281	7.2587	74.4926
CLAHE	8.2267	82.6101	10.5251	54.1432
GF&DDE	7.3881	74.8814	9.3054	78.9667
BEEPS&DDE	9.0576	90.6330	11.5466	83.7595
IE-CGAN	4.1403	43.2661	5.0793	74.3293
Proposed	9.6650	97.2950	12.2804	61.7801

**Table 6 sensors-23-08101-t006:** FPGA resource requirements.

Resource	Used	Available	% of All
BRAM_18K	408	1488	27
DSP48E	126	3528	3
FF	374600	682560	5
LUT	97685	341480	28

**Table 7 sensors-23-08101-t007:** Speed performance compared to PC implementation.

FPGA Maximum Clock Frequency	114 MHz
FPGA Maximum Frame Rate	147 fps
PC/MATLAB R2021a (i7-12700H @ 2.30 GHz)	5 fps
Speedup	29.4×

**Table 8 sensors-23-08101-t008:** Average metrics compared to PC implementation.

	AG	EI	FD	RMSC
PC	8.9718	92.4214	10.7972	61.8020
FPGA	8.0301	83.4944	9.5328	54.7012

## Data Availability

The data presented in this study are available on request from the corresponding author. The data are not publicly available due to privacy restrictions.

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
