# Peer review of "Multi-Scale FPGA-Based Infrared Image Enhancement by Using RGF and CLAHE"

_sensors, 2023, doi:10.3390/s23198101_

Round 1

Reviewer 1 Report

This paper proposes a method for FPGA-based infrared image enhancement.

The proposed method seems reasonable, and experimental results have verified the effectiveness of the method.

But the paper still has many limitations, and the authors are suggested to improve the paper from the following aspects.

First, the writing and presenting of the paper still need improvement.

Some writing, formatting, and language issues are still observed.

The authors are suggested to improve the paper from this aspect.

“…it may result in noise enhancement and the weakening of image details, thus affecting image quality…” - - - > Some surveys for image quality are suggested to be given here, e.g., Perceptual image quality assessment: a survey; Screen content quality assessment: overview, benchmark, and beyond.

As discussed in many studies, e.g., Study of subjective and objective quality assessment of audio-visual signals, Fixation prediction through multimodal analysis, A multimodal saliency model for videos with high audio-visual correspondence, integrating information of multiple modalities could benefit many tasks. The authors are suggested to give some discussions on this aspect and the above works, for example, is it possible to integrating both in infrared and visible images for this task?

The authors utilize a multi-scale strategy in the proposed method. Multi-scale processing has been widely used in the related applications, e.g., multi-scale processing for image quality assessment in: Objective quality evaluation of dehazed images, Quality evaluation of image dehazing methods using synthetic hazy images, A metric for light field reconstruction, compression, and display quality evaluation. The authors are suggested to give some introductions on this aspect and the above works.

Subjective & Objective Analysis

This part tries to evaluate the quality of the enhanced images, which is very related to image quality assessment. More introductions and representative image quality measures are suggested to be introduced here, e.g., BPRI proposed in Blind image quality estimation via distortion aggravation, BMPRI proposed in Blind quality assessment based on pseudo-reference image, and UCA proposed in Unified blind quality assessment of compressed natural, graphic, and screen content images,

All reference items are suggested to be double-checked. The formats should be unified.

None

Reviewer 2 Report

This paper presents a novel approach to address challenges associated with infrared imagery, such as low contrast and indistinct textures, caused by the long wavelength of infrared radiation and susceptibility to interference. The aim of this work is to improve the visual quality of infrared images in real-time by proposing a multi-scale FPGA-based method that leverages the rolling guidance filter (RGF) and contrast-limited adaptive histogram equalization (CLAHE). The work is divided into two parts: the first part focuses on the algorithmic aspects, while the second part addresses the implementation. The work is very interesting, the content is quite rich, and the structure is well-organized. However, I have some suggestions for the authors to improve the overall content of the work:

*Part 1: proposed Method (Algorithmic parts)

1- The choice of utilizing a Gaussian filter for the purpose of detail removal is under scrutiny due to two primary considerations. Firstly, the Gaussian filter's can potentially compromise the image's sharpness and the preservation of fine edge details, attributes that may be necessary for subsequent stages of image processing. Secondly, Gaussian filters can entail a very laborious computation when compared to alternative filters, such as the median filter, which is presumed to exhibit greater efficiency within this context. As a result, it would be highly beneficial if the authors were to employ both filters and subsequently make their selection based on the filter's performance relative to their specific objectives in image processing. This empirical evaluation stands to facilitate a more informed decision between the two filters, tailored to their unique application requirements.

2- In today's context, deep learning methods, such as Convolutional Neural Networks (CNNs), are extensively employed for enhancing infrared images. It raises the question as to why the authors didn't opt for a CNN-based approach. If their proposed method proves to be more effective, I would recommend that the authors conduct comparisons with deep learning-based techniques for a comprehensive evaluation.

3- In consideration of Table 4, it is evident that the Root Mean Square Contrast (RMSC) values obtained through the application of the proposed method to various scenes are significantly lower when compared to alternative methods. Consequently, the resulting images exhibit reduced contrast. What reasoning do the authors offer to explain their choice of the proposed method?

4- Page 12, Line 321: Authors need to verify the table number (Table 3 instead of Table 2).

5- Page 12, Line 327: Authors need to verify the table number (Table 4 instead of Table 3).

6- The authors are invited to correct typos and spelling errors throughout the entire article.

*Part 2: Hardware implementation 

7 - When using the HLS tool, it is not possible to impose a fixed clock frequency; therefore, the authors must provide the maximum clock frequency achieved by the architecture during its actual functionning.

8- The authors need to properly analyze the hardware resource utilization results for the FPGA as presented in Table 6. Specifically, they should explain why, for this architecture, the percentages of BRAM_18K and LUT utilization are high when compared to the utilization of DSP48E and FF resources.

*General recommendations

9- The "Conclusion" section is missing in this article; the authors should add this section.

10- The authors should enhance the "References" section with more recent citations in the same field.

Round 2

Reviewer 1 Report

Most of the concerns have been addressed.

None

Reviewer 2 Report

All the recommendations have been taken into consideration.